# Health Effects of Person-Under-Train Incidents on Train Drivers—A Systematic Review

**DOI:** 10.3390/healthcare13030248

**Published:** 2025-01-26

**Authors:** Johannes Lay, Andrea Kaifie

**Affiliations:** 1Institute for Occupational, Social, and Environmental Medicine, Medical Faculty, RWTH Aachen, 52074 Aachen, Germany; johannes.lay@rwth-aachen.de; 2Institute and Outpatient Unit for Occupational, Social, and Environmental Medicine, FAU Erlangen-Nuremberg, 91054 Erlangen, Germany

**Keywords:** person-under-train incident, post-traumatic stress disorder (PTSD), trauma, sickness absence, mental health, systematic review

## Abstract

**Background/Objectives:** A person-under-train (PUT) incident is a traumatic event for train drivers that can result in serious health consequences. This systematic review aimed to identify and synthesize evidence on these health effects, including post-traumatic stress disorder (PTSD), sickness absence (SA), and other related mental health outcomes. **Methods:** Following the PECO scheme, we searched the PubMed, Web of Science, and Embase databases for studies published between 1980 and 2021. We included cross-sectional, case–control, prospective, and interventional studies focusing on train drivers exposed to PUT incidents. Data synthesis followed PRISMA guidelines, with quality appraisal using the Downs and Black checklist. **Results:** The search was conducted on 22 September 2022, resulting in 3673 records. Nine studies were included, involving a total of 3425 participants. PTSD prevalence ranged from 0% to 55.3%, depending on study design and timing of data collection. Risk factors included repeated exposure, lack of social support, and proximity to the victim. SA varied widely, with durations ranging from 1 day to over 6 months. Drivers frequently reported depression, somatoform disorders, and other long-term psychological impacts. **Conclusions:** PUT incidents significantly affect the mental health of train drivers and their ability to return to work, frequently leading to extended sickness absence. The timely identification of at-risk individuals and the implementation of best-practice interventions could mitigate long-term effects. Future research should explore standardized follow-up periods to improve comparability.

## 1. Introduction

A sudden, unexpected event involving serious injury or death can be very traumatic for witnesses and can have lasting negative consequences, such as post-traumatic stress disorder (PTSD) [1]. People who are involved in or witness such an event, such as police or firefighters, are at an increased risk of developing PTSD or depression [2,3]. In addition to the trauma itself, the witness’s resilience and coping strategies are important in dealing with the trauma [4].

Train drivers are at a significant risk of experiencing a railway suicide or person-under-train (PUT) incident. In contrast to other occupational groups such as police officers or firefighters, train drivers are rarely physically endangered when they roll over or hit a person with the train, unless other trains or vehicles are involved in the collision or derailments occur. In addition, the incident occurs without warning; they often have no way of preventing it, are forced to witness it, and may feel responsible for it. Data from the European Union Agency for Railways show that PUT incidents are not uncommon. In 2022 alone, 2393 suicides on railway property [5] and 847 accidents involving people caused by rolling stock in motion [6] were reported. A study on London Underground drivers showed a prevalence of PTSD of 16.3% after PUT incidents [7]. This underlines a need to assess the potential health effects of PUT incidents for train drivers.

In 2015, almost 10 years ago, two systematic reviews already dealt with potential health effects in train drivers after a critical event [8,9]. However, since 2015, new scientific articles have been published and also more awareness of mental health disorders has arisen. The aim of this systematic review was to identify new studies that address health effects in train drivers following PUT incidents, to summarize their main findings, and to identify the most common and most serious health effects. We also focused on the impact of repeated PUT incidents in terms of habituation and sensitization. Our approach was to add to the work of the existing systematic literature and, where possible, present differences to the reviews and the more recently published work.

## 2. Materials and Methods

### 2.1. Protocol and Registration

This systematic review adhered to the guidelines of Preferred Reporting Items for Systematic Reviews and Meta-Analyses (PRISMA) and included a study protocol that was registered with the University of York’s International Prospective Register of Systematic Reviews (PROSPERO). On 22 August 2022, we registered our systematic review under the PROSPERO ID CRD42022355127.

### 2.2. Eligibility Criteria and Search Strategy

The main inclusion criterion was exposure to at least one PUT incident or railway suicide. We included case–control studies, prospective studies, observational studies, cross-sectional studies, and interventional studies. Outcomes included mental disorders or diseases, symptoms and other health effects, impacts on long-term health and occupational disability, and impacts on morbidity and mortality. Only studies that were published in English or German between 1980 and 2021 were included, as we believe that this time period should have included all relevant papers. Excluded were case reports, methodological studies, conference papers, letters to editors, and expert opinions. The inclusion and exclusion criteria can be found in detail in the study protocol on the PROSPERO website (https://www.crd.york.ac.uk/prospero/display_record.php?RecordID=355127, accessed on 9 December 2024).

A search term was developed according to the PECO scheme (Table 1). To ensure that the inclusion criteria specifically targeted train drivers, we categorized the population into “Worker” and “Profession”. This allowed for the precise filtering of studies and ensured that only those directly involving train drivers were included. Exposure referred to the experience of witnessing or being involved in a PUT incident. While no direct comparison groups were utilized, differences in exposure levels (e.g., single vs. multiple incidents) were considered. The outcomes focused on health effects such as PTSD, sickness absence, depression, and other health effects. The review was conducted in the electronic databases Embase, PubMed, and Web of Science on 22 September 2022. The detailed search terms can be found in Appendix A, Table A1.

### 2.3. Screening Process and Data Extraction

The screening process for this systematic review was carried out in three stages: title screening, abstract screening, and full-text screening. The process was designed to ensure a rigorous and unbiased selection of studies that met the predefined inclusion criteria. If no consensus could be reached between the reviewers in any stage, a third person was consulted.

In the initial stage, the titles of all identified records were screened independently by two reviewers. The primary objective in this stage was to exclude studies that were clearly unrelated to the research focus, such as those that did not involve train drivers, PUT incidents, or the health outcomes of interest. After title screening, the abstracts of the remaining studies were independently reviewed by the same two reviewers. This stage further refined the selection by assessing whether the studies focused on train drivers exposed to PUT incidents and the specified outcomes. The final stage involved a detailed screening of the full texts of studies that passed the abstract screening. The two reviewers independently assessed each study against the inclusion and exclusion criteria, considering aspects such as the study design, population, and reported outcomes. Studies that fully met the eligibility criteria were included in the review.

For data extraction, a table was generated, including the year of publication, author, title, study type/location, population/control, exposure/intervention, methods, outcome, and results. The study protocol intended to conduct a meta-analysis of the collected data where possible, but this was not feasible due to the poor comparability of the collected data.

### 2.4. Bias Assessment

For each original study, a bias assessment was independently conducted by the two reviewers. A bias assessment of internal validity, with two categories (bias and confounder), was conducted using a checklist published by Downs and Black [10]. Performance bias, detection bias, attrition bias, and reporting bias were adopted and modified from Higgins et al. [11]. The questions of the assessment were answered with “yes”, “no”, and “n.a.” (not applicable). If any question indicated a possible high risk in the individual assessment, the bias was considered high-risk. After both reviewers completed their assessment, they discussed disagreements until a consensus was reached, if necessary, by consulting a third reviewer from the institute, though this was not necessary. Once the bias assessment was completed, a table was created, categorizing each study as low-risk or high-risk for each type of bias.

## 3. Results

A total of 3673 papers were identified across all databases, including 778 in PubMed, 705 in Embase, and 2190 in Web of Science. One reviewer (J.L.) imported the 3673 papers into EndNote 20^©^. After removing 810 duplicates, 2863 papers were retained for title screening. The title screening was independently conducted by two reviewers (J.L., A.K.). During the title screening, many studies were excluded because their titles clearly did not meet the inclusion criteria. Most exclusions were carried out since the studies’ focus was unrelated to train drivers or person-under-train incidents. After completing the title screening, J.L. identified discrepancies. Discrepancies were resolved through discussion between the two reviewers. During the title screening, 2723 papers were excluded for not meeting the inclusion criteria, leaving 140 for abstract screening. The abstract screening excluded 99 papers, with one additional duplicate identified and removed. Forty papers progressed to full-text screening, of which nine were included in the systematic review.

Thirty-one papers were excluded: four due to the study population, one due to the outcome, five due to exposure, two due to the context, three due to the study design, two due to the studies being systematic reviews, and fourteen because they were already part of the two systematic reviews (Figure 1).

The review included three cross-sectional studies [12,13,14], three retrospective studies [15,16,17], two interventional studies [18,19], and one prospective study [20]. Six studies were conducted in Europe [12,15,16,17,19,20], two in Asia [13,14], and one in North America [18]. The studies collectively included 3425 participants. The characteristics of the included studies are summarized in Table 2. Additional details on specific data are provided in Appendix A, Table A2.

### 3.1. PTSD After PUT Incident/Rail Suicide

#### 3.1.1. Prevalence

The prevalence rates of PTSD after PUT incidents were described in seven papers. In one study [12], 84.8% exhibited subclinical symptoms, not severe enough to meet the diagnostic criteria for PTSD. A total of 6.6% had symptoms over the Impact of Event Scale (IES) cut-off score, indicating a probable PTSD diagnosis, though the authors noted that symptom severity remained low. One study [13] compared the lifetime prevalence of PTSD among subway drivers exposed to PUT incidents to those without such experiences. They reported a non-significant odds ratio (OR) of 2.06 (95% CI: 0.94–4.55) for developing PTSD. Additionally, the study compared subway drivers with at least two PUT incidents to those without any PUT experience. The OR was statistically significant at 3.57 (95% CI: 1.32–3.65) for developing PTSD.

Clarner et al. [15] reported a prevalence rate of 8.5% of trauma sequelae among their exposed population. Using an interventional study design, Bender et al. [18] investigated the difference in post-traumatic treatment. Four weeks post-incident, 38.7% of participants in the control group were diagnosed with PTSD, compared to 55.3% in the interventional group (*p* = 0.04). Both groups exhibited a significant reduction in symptoms over 1 to 3 months and 3 to 6 months (*p* < 0.001).

A prospective study reported that 48% (ICD-10) and 18% (DSM-IV) of the study population were diagnosed with PTSD one month after a railway suicide [20].

#### 3.1.2. Related Factors

Clarner et al. demonstrated that the duration of trauma sequelae was significantly correlated with the severity of the third-party injury (*p* = 0.024) [15]. They further demonstrated that the impact of the third-party injury severity was more evident when specific correlations were examined. A PUT incident where the person was injured or killed did not show a significant correlation with PTSD (*p* = 0.153). However, a strong correlation (*p* = 0.001) was observed when the person was at least seriously injured or killed. In this group, fewer drivers were included, but the accidents were more severe.

### 3.2. Sickness Absence (SA) After PUT Incident/Rail Suicide

#### 3.2.1. Days Off

Clarner et al. reported a median SA of 3 days (Q1–Q3 = 4–16). Drivers without trauma sequelae had a median SA of 1 day (Q1–Q3 = 0–3). Those with short trauma sequelae had a median SA of 4.5 days (Q1–Q3 = 1.25–9.75), and those with long trauma sequelae had a median SA of 41 days (Q1–Q3 = 21–54.5) [15].

The interventional study by Bender et al. reported a median return-to-work (RTW) time of 18 days in the control group (SD = 9.53; 95%CI = 0.00–36.67) and 44 days in the intervention group (SD = 37.12; 95% CI = 0.00–116.76) [18].

A study on London subway drivers reported an average SA duration of 47 days (95% CI = 38.0–55.8) [16]. Only 7% of drivers took no time off. When a third-party fatality occurred, 91% of drivers took SA with a median duration of 76 days.

Giupponi et al. observed that the average SA following a suicide or attempted suicide was 14 days. Of the drivers, 32% took 1–7 days off, 28% took 8–14 days, 26% took 15 days to 6 months, and 14% took longer than 6 months. Of the drivers who took more than 6 months off, 57% transitioned to indoor roles, while 43% prematurely retired [20].

#### 3.2.2. Related Factors

Bender et al. reported that experiencing a suicide (HR = 0.621; *p* = 0.049) and a higher burden of PTSD symptoms (HR = 0.344; *p* < 0.001) were significantly correlated with the duration of the RTW time [18]. Clarner et al. demonstrated in their retrospective study that the duration of SA increased with the severity of the potential traumatic event (PTE). The duration of SA increased by 46% (95% CI = 21–66) for a minor injury, up to 246% (95% CI = 74–600) for a severe injury, and up to 191% (95% CI = 39–534) for a fatal incident [17]. Chavda reported a significant relationship between the duration of SA and severe third-party injury or death (*p* < 0.01) [16].

Giupponi et al. observed statistically significant positive correlations between SA duration and mental health scores. Positive correlations were identified between SA duration and scores such as SOMS-2 (r = 0.692; *p* = 0.001) and SOMS-7 (r = 0.630; *p* = 0.001). PTSD (r = 0.446; *p* = 0.001), major depression (r = 0.799; *p* = 0.001), and an acute stress reaction (measured by IES; r = 0.571; *p* = 0.001) were correlated with SA duration [20].

### 3.3. Habituation/Sensitization of PUT Incidents/Railway Suicide

Doroga and Baban reported a significant correlation between an increasing number of PUT incidents and lower PTSD severity (*p* < 0.01) [19]. In 2013, they demonstrated that total PTSD symptom scores significantly decreased (F = 5.14; *p* = 0.00) with the number of incidents, as did other IES-R (Impact of Event Scale—Revised) symptom groups, including intrusion (F = 3.37; *p* = 0.03), avoidance (F = 5.13; *p* = 0.00), and hyperarousal (F = 3.28; *p* = 0.04). Drivers with one to two incidents had a mean total IES-R score of 20.73 (SD = 8.86), those with two to five incidents scored 16.01 (SD = 10.30), and drivers with six or more incidents scored 14.16 (SD = 9.40) [12]. A score of 33 points or higher represents the optimal cut-off value for PTSD [21].

Kim et al. reported a significantly increased odds ratio (OR = 3.57; 95% CI = 1.32–3.65) for the lifetime prevalence of PTSD among drivers exposed to two or more PUT incidents [13]. A prospective study found that all drivers exposed to four or more railway suicides developed PTSD. The study also reported that “positive associations were found for the number of critical incidents in service (r = 0.314, *p* = 0.14)” [20].

Regarding SA, drivers with a previous history of PUT incidents took significantly more time off after an incident than those without such a history, when stratified by incident type (*p* < 0.01) [16].

### 3.4. Other Effects of PUT Incidents/Railway Suicide

Giupponi et al. [20] reported in their prospective study prevalence rates of 30% for major depression, 20% for somatoform disorders, 8% for substance-related disorders, and 8% for a borderline personality disorder among subway drivers one month after witnessing a railway suicide.

### 3.5. Bias Assessment

A summary of the bias assessment is provided in Table 3. The categories “internal validity—confounding” and “selection bias” had the highest proportion of “high risk” with 33% each. Additionally, “internal validity—bias” and “performance bias” had each a 22% “high risk”, while “detection bias” accounted for 11%. Four out of the nine papers were classified as entirely low-risk. The highest proportion of high risk observed in a single paper was in four out of the seven categories [12].

## 4. Discussion

This systematic review aimed to summarize the current scientific knowledge on the health effects of PUT incidents/rail suicide on train drivers and to evaluate new data. Nine papers were included in our analyses with a strong focus on PTSD and SA. For some drivers, a PUT event resulted in severe mental health consequences, ranging from flashbacks or mental disorders to incapacity and early retirement.

The prevalence rate of PTSD varied considerably across the included studies, ranging from 0% to 55.3%. This variability can be attributed to differences in the study design and the timing of data collection after the PUT incident. Cross-sectional and retrospective studies assessed prevalence rates irrespective of the time elapsed since the event. In the interventional and observational studies, prevalence data were collected at specific intervals, often shortly after the event. PTSD prevalence rates decreased as data collection occurred later after the event. Especially, the interventional study by Bender et al. showed that the prevalence of PTSD symptoms, measured by the MPSS, decreased significantly over time [18]. This trend was also observed in the already-existing systematic reviews of Bardon et al. as well as Clarner et al. [8,9].

A significant risk factor for trauma sequelae, including PTSD and SA, is whether a third party was seriously injured or killed during the PUT incident. An umbrella review and meta-analysis identified witnessing an injury or death as a significant risk factor for developing PTSD [1], consistent with the findings of Clarner et al. [15]. In terms of sickness absence, Clarner et al. demonstrated that the duration of sickness absence increased with the severity of the PTE, which was measured by third-party injury [17]. This is intuitive, as the severity of the PTE is a known risk factor for trauma sequelae, and sickness absence logically increases in the context of such sequelae and mental disorders.

A promising approach to mitigate these negative health effects is the establishment of a standardized and evidence-based follow-up for potentially traumatic incidents in order to detect mental health disorders or diseases early and to initiate sufficient therapy. The prospective study by Giupponi et al. [20] demonstrated a significant correlation between the combination of standardized questionnaires and the detection of PTSD. These questionnaires included the Cologne Trauma Inventory (KTI), Screening for Somatoform Symptoms (SOMS), and the Impact of Event Scale (IES). The IES assesses intrusive and dissociative/avoidance symptoms in the acute aftermath of an event. The KTI evaluates previous traumatic experiences from childhood and adulthood, offering a prognostic criterion for PTSD risk following current trauma exposure. SOMS-2 evaluates medically unexplained symptoms occurring within the two years prior to the traumatic event, while SOMS-7 assesses new physical symptoms arising within the past seven days. A combination of these questionnaires could effectively assess PTSD risk among affected train drivers and guide more intensive support for those at the highest risk of developing PTSD. However, whether this approach can prevent PTSD development or positively impact follow-up care requires further investigation.

It remains unclear whether train drivers habituate to or become sensitized by PUT incidents. Two studies [12,19] reported habituation, and four [9,13,16,20] described sensitization. In the systematic review of Bardon et al., both effects were identified [8]. Considerable evidence suggests that repeated exposure does not lead to habituation, as observed in our analyses. Additionally, studies with a similar scope report that repeated exposure to traumatic events tends to result in increased psychological symptoms or problems [2,3,22,23]. Moreover, whether other factors influence habituation or sensitization following traumatic events should be investigated to better understand this mechanism.

Compared to the already-existing systematic reviews, the prevalence rates of PTSD differed significantly between our study and the other two systematic reviews and is explainable by the variability of prevalence rates among the included individual studies. However, all three reviews observed a decrease in PTSD prevalence over time. Similarly to the findings of the two previous reviews, the critical role of pre-traumatic resilience as a major factor influencing psychological outcomes is of importance. Giupponi et al. showed in their prospective study that pre-traumatic health status was significantly associated with PTSD outcomes. This finding highlights the importance of preventive measures and resilience-building efforts but also acknowledges that not all risk factors can be mitigated. Interestingly, this review reported significantly longer sickness absences (3–76 days) compared to the previous two reviews (3–19 days), further illustrating the severe health consequences of PUT incidents for train drivers. In addition, this review demonstrated the uncertainty about whether habituation or sensitization occurs among train drivers after several traumatic events, although existing evidence and the broader trauma literature suggest that sensitization is more likely. These findings reinforce the need for further research to deepen the understanding of these psychological responses and to optimize prevention and intervention strategies.

Like most systematic reviews, this study has its limitations. This study drew data from nine different papers with varying study designs. The search was restricted to studies in German or English, conducted between 1980 and 2022, and indexed in PubMed, Web of Science, and Embase. Consequently, studies in other languages, indexed in different databases, or conducted outside this time frame were excluded from the analysis. The data collection methods varied across the studies. PTSD diagnoses varied, with some studies relying on medical diagnoses and others using the IES-R to assess symptoms and assign participants to a cohort based on cut-off scores. This approach allowed authors to assign participants to a PTSD cohort, despite the fact that PTSD can only be formally diagnosed 4 weeks post-event.

Another limitation is the short and variable follow-up period of the studies. The follow-up in the prospective studies typically lasted up to a maximum of 6 months [18,20], whereas the systematic reviews of Bardon et al. and Clarner et al. [8,9] indicated prevalence changes beyond this period. Subjective evaluations, including anger, fear, and disappointment, were described but are challenging to measure or compare objectively. Furthermore, the nine studies were conducted in different countries and cultural contexts, each with distinct approaches to dealing with death, suicide, and coping after traumatic events.

The bias assessment revealed a high proportion of high-risk ratings in the selection bias and internal validity confounders categories. This was primarily due to many retrospective studies relying on voluntarily data provided by railway companies. This represents a significant source of bias. Future research should ensure that data collection is conducted independently of railway companies. For instance, an Australian study reported that train drivers misreported information during mandatory health screenings due to fear of negative consequences [24].

## 5. Conclusions

In summary, a PUT incident is a traumatic experience for professional drivers. Many of them experience short-term mental health challenges, including acute stress reactions. This is reflected in drivers’ sickness absences, even when no long-term psychological disorder is diagnosed. A small proportion of drivers develop PTSD and/or other psychological disorders, such as depression or substance abuse. This is associated with a significantly increased number of sick days.

Drivers should be protected from the effects of PTSD and it should be identified in an early stage of the condition. It is essential to identify train drivers at a high risk of developing a stress disorder following a PUT incident and to implement best-practice interventions. However, the best methods for identifying these drivers and designing effective interventions remain unclear and require further research.

## Figures and Tables

**Figure 1 healthcare-13-00248-f001:**
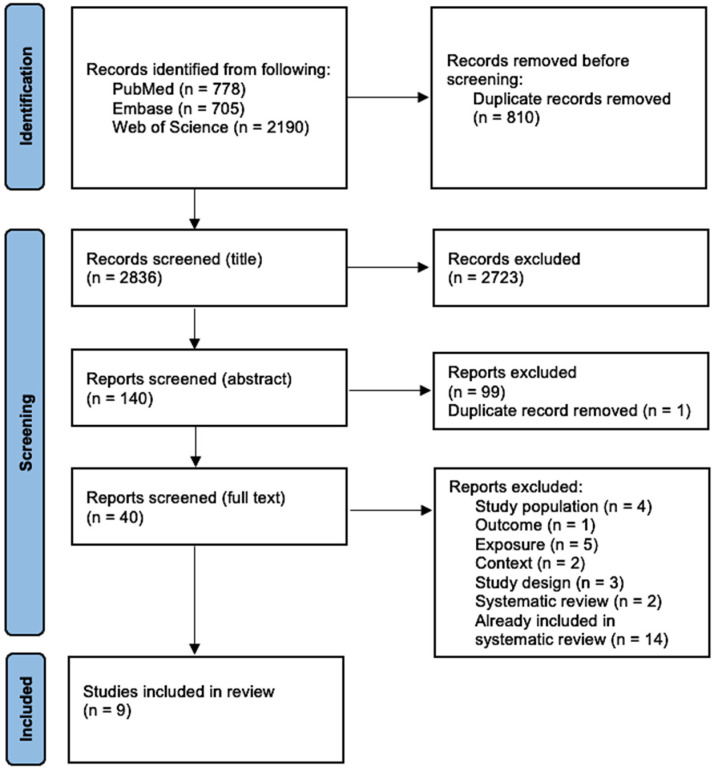
PRISMA flowchart of screening process.

**Table 1 healthcare-13-00248-t001:** PECO scheme for used search term.

PECO Scheme			
Worker	Profession	Exposure	Outcome
Driver	Train	Death	Health
Worker	Subway	Suicide	Disorder
	Underground	Person-under-train	PTSD
	Tram	(PUT)	Symptoms
	Locomotive	incident	Disability
	Rail	Accident	Mental health disorder
	Railroad	Collision	Panic attacks
	Railway		Job change
	Transport		Disease
	Public transport		Morbidity
			Mortality

**Table 2 healthcare-13-00248-t002:** Included studies and study characteristics [12,13,14,15,16,17,18,19,20].

Included Studies and Study Characteristics							
Year	Title	Author	Study Type/ Location	Population/ (Control Group)	Exposure/Intervention	**Methods**	**Outcome**
2013	Traumatic exposure and posttraumatic symptoms for train drivers involved in railway incidents	C. Doroga and A. Baban	Cross-sectional study Romania	n = 193 0 PUT: n = 41 (21.2%); 1–2 PUT: n = 38 (19.7%); 3–5 PUT: n = 65 (33.7%); >5 PUT: n = 49 (25.4%)	PUT	Questionnaire	IES-R; GHQ-28; intrusion; avoidance; hyperarousal
2014	The association between psychiatric disorders and work-related problems among subway drivers in Korea	S. E. Kim, H. R. Kim, J. I. Park, H. W. Lee, J. Lee, J. Byun, and H. W. Yim	Cross-sectional study South Korea	n = 980 Control group—national data 2011, male	PUT	Questionnaire	PTSD
2015	Psychotrauma after occupational accidents in public transportation. A pilot study to support concepts, influencing factors and occupational health-care	A. Clarner, J. Krahl, W. Uter, H. Drexler, and A. Martin	Retrospective study Germany	n = 59	Incident that required on-site trauma care (accident with person—51%; suicidal incident—19%; near-accident; accident with road vehicle; physical attack; other event)	Company data	PTSD; trauma sequelae; SA
2016	Best practice intervention for posttraumatic stress disorder among transit workers	A. Bender, R. Eynan, J. O’Grady, R. Nisenbaum, R. Shah, and P. S. Links	Interventional study Canada	n = 141 TAU: n = 62 BPI: n = 79	PTE (suicide—25%; accident—16%; physical or verbal assault; robbery; witnessed accident) TAU: treated as usual; consent to collect information from doctors and family BPI: specialized multidisciplinary psychological treatment program	Questionnaire; patient data	PTSD; risk factors; SA and risk factors
2016	Factors associated with suicide ideation among subway drivers in Korea	J. Byun, H. R. Kim, H. E. Lee, S. E. Kim, and J. Lee	Cross-sectional study South Korea	n = 980 SIG: n = 33 NSIG: n = 947	PUT	Questionnaire	Suicide-ideation group (SIG); no-suicide-ideation group (NSIG)
2016	Sickness absence of train drivers after track incidents	S. Chavda	Retrospective study U.K.	n = 687	Near miss of person on platform (n = 248) Near miss of person on track (n = 263) PUT with significant injury (n = 68) PUT with fatality (n = 106) Missing data (n = 8)	Company data	SA and risk factors
2017	Sickness absence among peer-supported drivers after occupational trauma	A. Clarner, W. Uter, L. Ruhmann, N. Wrenger, A. Martin and H. Drexler	Retrospective interventional study Germany	n = 259 VGF: n = 84 VAG-0: n = 80 VAG-1: n = 95	PTE (acc.—36%; collision—30%; suicide—7%; attack/conflict; fall of passenger; other type)	Company data; patient data	SA and risk factors
2019	Posttraumatic stress reactions of underground drivers after suicides by jumping to arriving trains; feasibility of an early stepped care outpatient intervention	G. Giupponi, H. Thoma, D. Lamis, A. Forte, M. Pompili, and H. P. Kapfhammer	Prospective observational study Germany	n = 50	Suicide	Questionnaire	PTSD; SA; CIDI; IES; SOMS-2; SOMS-7; KTI; SA and risk factors
2013	Impact of Work-Related Trauma on Acute Stress Response in Train Drivers	C. Doroga and A. Baban	Interventional study Romania	n = 76 0 PUT: n = 23; ≥1 PUT—low-level-PTSD-symptoms group: n = 26; ≥1 PUT—high-level-PTSD-symptoms group: n = 27	Group 1 (0 PUT)—simulated PUT Group 2 (≥1 PUT)—simulated PUT	Questionnaire; heart rate measure	IES-R; heart rate reactivity; dysphoric symptoms

PUT: person under train; PTSD: post-traumatic stress disorder; SA: sickness absence; GHQ-28: General Health Questionnaire-28; PTE: potential traumatic events; TAU: treatment as usual; BPI: best-practice interventions; SIG: suicide-ideation group; NSIG: no-suicide-ideation group; VGF: peer support at supervisor level; VAG-1: peer support by colleagues; VAG-0: non-intervention group; CIDI: Composite International Diagnostic Interview; IES: Impact of Event Scale; SOMS: Screening for Somatoform Disorders; KTI: Cologne Trauma Inventory.

**Table 3 healthcare-13-00248-t003:** Risk of bias assessment [12,13,14,15,16,17,18,19,20].

Bias Assessment							
Study Author (Year)	Internal Validity—Bias	Internal Validity—Confounder	Performance Bias	Detection Bias	Attrition Bias	Reporting Bias	Selection Bias
Doroga and Baban (2013)	high	high	low	high	low	low	high
Kim et al. (2014)	low	low	low	low	low	low	low
Clarner et al. (2015)	low	low	high	low	low	low	low
Bender et al. (2016)	high	high	low	low	low	low	high
Byun et al. (2016)	low	low	low	low	low	low	low
Chavda (2016)	low	low	low	low	low	low	low
Clarner et al. (2017)	low	low	high	low	low	low	low
Giupponi et al. (2019)	low	low	low	low	low	low	low
Doroga and Baban (2012)	low	high	low	low	low	low	high
High-risk count	2	3	2	1	0	0	3
Share of high risk	22%	33%	22%	11%	0	0	33%

## Data Availability

The datasets supporting the conclusions of this article are included within the article.

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
