# Peer review of "Health Effects of Person-Under-Train Incidents on Train Drivers—A Systematic Review"

_healthcare, 2025, doi:10.3390/healthcare13030248_

Round 1
Reviewer 1 Report
Comments and Suggestions for Authors
Dear Respectable Authors
Thank you for considering a great area of research related to trauma specifically in train drivers. You conducted a systematic review to identify and synthesize studies describing these consequences. I read your manuscript in-depth, and I think your results are interesting, but the way you report the manuscript needs some revisions. I hope these comments will promote the quality of your manuscript.
- Title, in my opinion, it is better to refine your title and select a better title representation of your aim of the study. Also, It is better to remove the title from the question form and only mention the type of study after the colon at the end.
- Abstract, based on journal guidelines, please remove all subheadings from the abstract section.
- Abstract, aim, please add a specific aim of your study. The aim you write here is not representative. Which consequences do you mean? Please add a clear aim and it is better to follow PICO format for stating your aim.
- Abstract, please add more details of the methods. Also, as the number of records are related to the results then you just remove the numbers from here and add them at the first statement of the results section. Please add the exact date of the search or search period, and add eligibility criteria in brief. Please add the checklist that you used for quality appraisal. Also, add more details regarding the methods of synthesizing the data.
- Abstract, please start your results with the final included studies, then, add information related to the prevalence of post-traumatic stress disorder (PTSD) and sickness absence (SA) among train drivers, along with contributing risk factors. Please add the exact results for the above-mentioned subject. What is the prevalence rate? What are the contributing risk factors? Etc.
- Abstract, refine your abstract after refining your aim and results. The conclusion must be a direct answer to the study question or aim of the study with some practical or future research recommendations.
- Keywords, please add full term for PTSD.
- Introduction section, please enrich your introduction. Remove the extra references. You have mentioned eight references for the first paragraph, which is too many and there is no need for so many references. Also, in one paragraph, review the literature on the prevalence of this phenomenon and the factors affecting it. Then, state the importance of the topic, the gaps in knowledge, and what knowledge gaps your results will cover. Finally, state the purpose of your study clearly at the end of the introduction by stating the necessity of such research.
- Methods, it is amazing to me that you have claimed that follow PRISMA but your method did not follow this checklist. The first item in the methods section after protocol and registration is eligibility criteria not search. You can not start a systematic review until the eligibility criteria not specified. Please add eligibility criteria, then, mention database and search. After that, you need to mention the selection process, quality appraisal, data items and data extraction, and method of synthesis.
- Line 58, please remove the aim from here and replace it at the end of the introduction section.
- Please add inclusion and exclusion criteria under the eligibility criteria following the PECO format. Also, you need to remove the number of records as well as Figure 1 from the methods section. Please add a full search strategy for all databases with the exact date of search and the number of records for each database. It is better to add this information as a supplementary file.
- Line 73, you stated that include a systematic review and meta-analysis in your study. This is quite wrong. We cannot include systematic review and meta-analysis in a systematic review. Please follow the Cochrane guidelines for this. When we include the systematic review and meta-analysis in a systematic review, in fact, we are subject to reporting bias and report more results than there are. Because these reviews and meta-analyses also use the results of original studies, combining these reviews with original studies at the same time is a form of bias. We incorporate systematic reviews and meta-analyses into other types of reviews, such as scoping reviews or umbrella reviews.
- Please make the requested corrections so that your article is eligible for further review. In this case, your article will not be accepted and the systematic reviews and meta-analyses will have to be removed from your results. After the results are corrected, the rest of the review process will be carried out.
Best regards,
Comments on the Quality of English LanguageThere are several punctuation and grammatical errors in the text. Please benefit from a native English speakers.
Reviewer 2 Report
Comments and Suggestions for Authors
This is an interesting and unique look of the literature for train drivers.
· The article title does not seem to reflect the themes that the articles in the review found.
· There are many small grammar and spelling errors through the paper (drive VS. driver) and the paper appears to not have been thoroughly reviewed before submission.
· There are crucial pieces of information missing in the process of the literature review -
o What criteria was used in the title screening process?
o Such a high number of titles were excluded (2723), what was the reason?
o What was the reason for the reports not retrieved?
· It can be appreciated that the reviewers created a table with the 11 reviewed articles. It would also strengthen and clarify the data by creating a table with the themes found.
· Overall without clear explanation of why 2723 articles were excluded the article review seems incomplete.
Comments on the Quality of English Language· There are many small grammar and spelling errors through the paper (drive VS. driver) and the paper appears to not have been thoroughly reviewed before submission.
Reviewer 3 Report
Comments and Suggestions for Authors
Dear author
I have reviewed the manuscript, It seems novel and interesting.
Your search strategy is a good one, with a fine PRISMA flowchart.
Everything is ok, but the title is ambiguous, try to change it.

Reviewer 4 Report
Comments and Suggestions for Authors
This is a well-researched attempt at a systematic review of the effects of trauma on the health of train drivers who have been involved in train accidents. However, I do think there are some things that need to be improved in terms of manuscript writing. The things that need to be supplemented in the submitted manuscript are as follows.
1. The introduction should be greatly improved. For example, you should describe in detail the traumatic experiences of train drivers who have experienced train accidents. And you need to describe in detail how previous studies on this subject have been conducted. And the reason for doing SR should be clear.
2. And, instead of just presenting the PECO scheme in a table and leaving it up to the reader to figure out, you should state it specifically in the text of the paper.
3. And it would be better if you presented the results of the bias assessment in a table or figure.
4. Please provide more specific clinical implications of your findings.
Round 2
Reviewer 1 Report
Comments and Suggestions for Authors
Dear Respectable Authors
Thank you for your clarification.
Cheers
Reviewer 2 Report
Comments and Suggestions for Authors
It appears that authors have sufficiently improved the manuscript according to reviewer's remarks.